New materials of plesiacerathere (Perissodactyla, Rhinocerotidae) from the late Early Miocene of Northern China

Sun Danhui 1 2
Deng Tao dengtao@ivpp.ac.cn 1 2
Wang Shiqi 2
1 University of Chinese Academy of Sciences , Beijing , China
2 Institute of Vertebrate Paleontology and Paleoanthropology, Chinese Academy of Sciences , Beijing , China
Moncunill-Solé Blanca
Electronic publication date: 2024 Jan 31
Publication date: 2024
Volume: 12
Electronic Location ID: e16822
Received 2023 Jul 12; Accepted 2024 Jan 2
Copyright: ©2024 Sun et al.
Copyright year: 2024
Copyright holder: Sun et al.
License: This is an open access article distributed under the terms of the Creative Commons Attribution License, which permits unrestricted use, distribution, reproduction and adaptation in any medium and for any purpose provided that it is properly attributed. For attribution, the original author(s), title, publication source (PeerJ) and either DOI or URL of the article must be cited.
License URL: https://creativecommons.org/licenses/by/4.0/

Keywords: Plesiaceratherium, Osteology, Phylogeny, Northern China

Funding: The Second Comprehensive Scientific Expedition on the Tibetan Plateau 2019QZKK0705 The National Natural Science Foundation of China 42302013 The National Key Research and Development Program of China 2023YFF0804501 This research was supported by the Second Comprehensive Scientific Expedition on the Tibetan Plateau (2019QZKK0705), the National Natural Science Foundation of China (42302013), and the National Key Research and Development Program of China (2023YFF0804501). The funders had no role in study design, data collection and analysis, decision to publish, or preparation of the manuscript.

==============================
As a member of Aceratheriinae, the genus Plesiaceratherium in Europe is widely distributed and highly diverse. However, only one species of Plesiaceratherium (i.e., P. gracile) exists in China with a discontinuous distribution range. Recently, we have discovered new materials of Plesiaceratherium in the lower layers of the Zhang’enbao Formation exposed in Miaoerling in Tongxin County, China. The new materials are well-preserved and can be separated from other Plesiaceratherium species by the following combination of features: the long and generally flat skull, with closed frontoparietal crests; the deep nasal notch at the level of P4; the high supraorbital margin, with its anterior margin at the level of the M1/M2 boundary; the medium-sized upper I1, with an oval abraded surface; the semi-molarized upper premolars with the protocone and hypocone joined by a lingual bridge; the strong constrictions of protocone on the upper molars; the absent buccal cingulum on upper cheek teeth; the cheek teeth are covered by cement on the buccal walls; the convex base of mandibular corpus; the inclined backward ramus; and the mandibular foramen above the teeth neck. Based on the combination of characteristics and phylogenetic analysis, we herein establish the new species as Plesiaceratherium tongxinense sp. nov. living in the late Early Miocene. Phylogenetic analysis reveals that P. tongxinense is in the basal position of the genus Plesiaceratherium, providing more detailed morphological characteristics of the plesiaceratheres.

Introduction

As a member of Aceratheriinae, the genus Plesiaceratherium is well-documented throughout Eurasia (Young, 1937; Yan & Heissig, 1986; Antoine, Bulot & Ginsburg, 2000; Peter, 2002; Antoine et al., 2010). Plesiaceratherium is a primitive acerathere rhinoceros with elongated nasals, long but robust limb bones, and a four-toed manus (Yan & Heissig, 1986; Antoine, 2002). Young (1937) established the genus Plesiaceratherium based on some isolated teeth and limb bones discovered in the Early Miocene of Shanwang in Linqu, Shandong Province, China, with P. gracile serving as the type species. Later, Chen & Wu (1976) described some dental materials from the Miocene of Jiulongkou in Cixian, Hebei Province, China, as belonging to P. gracile. Then, more well-preserved materials of Plesiaceratherium were discovered in Shanwang, China, including many skeletons, complete skulls, and many teeth and limb bones, giving more detailed characteristics of Plesiaceratherium (Yan, 1983; Yan & Heissig, 1986). In addition to the discovery of Plesiaceratherium in eastern China, Plesiaceratherium also has been found in the Early Miocene of Lunbori, Baingoin County, northern Tibet, China, including a humeral material (Deng et al., 2012). In Asia except for China, Fukuchi & Kawai (2011) described Plesiaceratherium sp. based on a right mandibular fragment with p2–m3 from the Lower Miocene Nakamura Formation in Japan.

In Europe, five species have been attributed to the genus Plesiaceratherium, including Plesiaceratherium fahlbuschi, Plesiaceratherium platyodon, Plesiaceratherium lumiarense, Plesiaceratherium aquitanicum, and Plesiaceratherium balkanicum. Heissig (1972) established the species Aceratherium fahlbuschi based on a nearly complete, uncrushed skull (BSPG 1959 II 400) as a holotype discovered in the locality Sandelzhausen in Bavaria, which was later classified as P. fahlbuschi by Yan (1983). Mermier (1895) established the species Aceratherium platyodon based on a deformed skull with mandible, afterwards Yan (1983) referred it to P. platyodon. Yan & Heissig (1986) established the species P. lumiarense based on the right maxilla with P1-M3 as a holotype from Portugal, which was previously identified as Aceratherium lumiarense by Antunes & Ginsburg (1983). Antoine & Becker (2013) referred a species established by Répelin (1917) to P. aquitanicum. Becker & Tissier (2020) established a new species P. balkanicum based on a left premolar row as holotype from Bugojno Basin, Bosnia-Herzegovina. While there was a species from the lower Miocene of Can Julia, Barcelona, Spain, referred to Plesiaceratherium mirallesi by Yan (1983), but Lu et al. (2016) considered that P. mirallesi should be excluded from the genus Plesiaceratherium, and the initial genus name Dromoceratherium should be revived. Besides, in Africa, there were two incomplete skulls from Nyakach, Kenya, numbered KNM-NC-10486 and KNM-NC-10510, provisionally referred to Plesiaceratherium sp. by Geraads (2010).

Until now, the genus Plesiaceratherium in Europe is widely distributed and highly diverse, but only one species of Plesiaceratherium (i.e., P. gracile) exists in China with stratigraphically discontinuous distribution range. Fortunately, we have discovered new materials of Plesiaceratherium in Tongxin County, Ningxia Hui Autonomous Region, China. The Tongxin region contains an abundance and continuous deposit of Cenozoic sediments (Wang et al., 2011; Wang et al., 2016). Our new plesiacerathere materials reported here were found in the lower layers of the Zhang’enbao Formation exposed in Miaoerling, which dates to the late Early Miocene (Wang et al., 2016). The studied materials allow the description of a new species of plesiacerathere, Plesiaceratherium tongxinense sp. nov. providing more detailed multiple characters of Plesiaceratherium.

Materials & Methods

The studied fossils are an adult skull and a mandible discovered in Tongxin, Ningxia Hui Autonomous Region, China, and stored in the collection of the Institute of Vertebrate Paleontology and Paleoanthropology (IVPP), Chinese Academy of Sciences, Beijing, China. The fossils are described and identified through anatomical descriptions, comparative anatomy as well as biometrical measurements. Rhinocerotid terminology and taxonomy follow Heissig (1972), Heissig (1999), Guérin (1980) and Antoine (2002). Anatomical features described follow the same sequence as in Antoine (2002), and Antoine et al. (2010). The specimens were measured by the procedures described in Guérin (1980).

Phylogenetic analysis

The phylogenetic analysis in this paper is performed using a modified data matrix from Antoine (2002) and Antoine (2003) to assess the phylogenetic position of the new specimen which can be found in the Appendix section. There are 282 morphological features in the matrix under analysis in this work. The 39 taxa that make up the current matrix are all species-level coded. A heuristic search was used to perform the phylogenetic analysis using PAUP4.0a169 (Swofford, 2002), with TBR, 1,000 replications with random addition sequence, 10 trees held at each step, treating gaps as missing, and no differential weighting or topological constraint a priori. Apart from characters 72, 94, 102, 140, and 187 (which are unordered), all multistate characters were considered as ordered.

Nomenclatural acts

The electronic version of this article in Portable Document Format (PDF) will represent a published work according to the International Commission on Zoological Nomenclature (ICZN), and hence the new names contained in the electronic version are effectively published under that Code from the electronic edition alone. This published work and the nomenclatural acts it contains have been registered in ZooBank, the online registration system for the ICZN. The ZooBank LSIDs (Life Science Identifiers) can be resolved, and the associated information viewed through any standard web browser by appending the LSID to the prefix http://zoobank.org/. The LSID for this publication is: urn:lsid:zoobank.org:pub:D79AA940-F686-4CD7-A751-C94CC0E30E44. The online version of this work is archived and available from the following digital repositories: PeerJ, PubMed Central SCIE and CLOCKSS.

Results

Systematic paleontology

Order Perissodactyla Owen, 1848	
Family Rhinocerotidae Gray, 1821	
Subfamily Aceratheriinae Dollo, 1885	
Tribe Aceratheriini Dollo, 1885	
Genus PlesiaceratheriumYoung, 1937	

Type species. Plesiaceratherium gracile Young, 1937

Other species. P. platyodon (Mermier, 1895), P. aquitanicum (Répelin, 1917), P. fahlbuschi (Heissig, 1972), P. lumiarense (Antunes & Ginsburg, 1983), P. naricum (Pilgrim, 1910), P. balkanicum Becker & Tissier, 2020, and P. tongxinense sp. nov.

Revised Diagnosis. Medium to large-sized primitive acerathere; limb bones more slender than in other Miocene aceratheriine genera; the nasal bones are elongated and straight, with a deep nasal notch at the level of P4; I1 is medium-sized, i2 is large and slightly curved; the upper cheek teeth have low crowns; the upper premolars are semi-molarized; the lower premolars are narrow and long, with relatively shallow ectoflexid (Yan & Heissig, 1986; Lu et al., 2016).

Distribution. Early Miocene (MN 1-5), Eurasia.

Plesiaceratherium tongxinense sp. nov.	
(Figs. 1–4; Tables 1–4)	

Holotype. IVPP V 23959, a well-preserved and complete skull and mandible (Figs. 1–3) representing a full adult individual, which are preserved at the Institute of Vertebrate Paleontology and Paleoanthropology, Chinese Academy of Sciences, Beijing, China.

Derivation of name. The specific name, tongxinense, refers to the geographical location of the discovery.

Type locality and horizon. Miaoerling in Shishi Township, Tongxin County, Ningxia Hui Autonomous Region, China; late Early Miocene.

Diagnosis. The skull is long and relatively flat, with closed frontoparietal crests; the supraorbital margin is high and its anterior margin is located at the level of the M1/M2 boundary; the upper I1 is developed and specialized; the cheek teeth are covered by cement on the buccal walls; the protocone on the upper molars has developed anterior and posterior constrictions; the buccal cingulum is absent on upper cheek teeth; the base of mandibular is convex; the ramus is inclined backward; and the mandibular foramen is located above the teeth neck.

Figure 1 Photographs and schematic illustrations of the skull of Plesiaceratherium tongxinense sp. nov., holotype (IVPP V23959).

(A) Dorsal view; (B) lateral view; (C) ventral view.

Figure 2 Photograph of the posterior view of the skull of Plesiaceratherium tongxinense sp. nov., holotype (IVPP V23959).

Figure 3 Photographs and schematic illustrations of the skull of Plesiaceratherium tongxinense sp.nov., holotype (IVPP V23959).

(A) Lateral view; (B) occlusal view.

Figure 4 Strict consensus of four most parsimonious trees, with 1,233 steps in PAUP (consistency index = 0. 3001; retention index = 0.5738), showing systematic position of Plesiaceratherium tongxinense sp. nov.

The clade (Node A) includes the members of Teleoceratina supported by thirty-one equivocal synapomorphies. The clade (Node B) includes the plesiaceratheres supported by twenty equivocal synapomorphies. The clade (Node C) containing elasmotheres is supported by forty equivocal synapomorphies.

Table 1 Measurements of the cheek teeth of Plesiaceratherium tongxinense sp. nov. and comparison with other Plesiaceratherium species.

(mm) (Measurements of P. gracile, P. fahlbuschi, P. platyodon, and P. mirallesi are from Yan & Heissig (1986); P. lumiarense is from Antunes & Ginsburg (1983) and Ginsburg & Bulot (1984)).

Teeth	P. tongxinense sp. nov.	P. gracile	P. fahlbuschi	P. platyodon	P. mirallesi	P. lumiarense	P. aquitanicum	P. balkanicum	
DP1	L	–	–	–	20	23	23	–	21	
	W	–	–	–	18	20	19	–	20	
P2	L	28	31	24	30	29	29	29	29	
	W	36	36	33	35	36	35	36	37	
P3	L	34	34	29	28	35	35	34	34	
	W	45	42	39	43	45	44	46	43	
P4	L	39	37	32	34	38	38	36	35	
	W	51	43	43	45	49	49	51	47	
M1	L	46	46	32	35	42	45	–	34	
	W	53	48	44	43	47	49	–	41	
M2	L	54	48	36	38	46	45	47	38	
	W	56	50	47	46	51	50	53	45	
M3	L	44	43	41	37	42	38	42	33–36	
	W	52	45	42	42	45	49	49	37–40	
p2	L	23	28	26	27	32	–	–	–	
	W	16	19	16	17	14	–	–	–	
p3	L	29	31	30	33	35	–	–	–	
	W	23	20	19	23	21	–	–	–	
p4	L	37	35	32	33	39	–	–	–	
	W	27	26	22	24	25	–	–	–	
m1	L	38	38	32	35	43	–	–	–	
	W	28	28	22	24	23	–	–	–	
m2	L	44	40	37	38	46	42	–	36	
	W	29	26	22	24	24	29	–	24	
m3	L	44	43	–	40	47	43	–	39	
	W	27	27	–	23	25	27	–	24	

Table 2 Measurements of the skull of Plesiaceratherium tongxinense sp. nov. (IVPP V 23959).

(mm) (Measurements of P. gracile, P. fahlbuschi, P. platyodon, and P. mirallesi are from Yan & Heissig (1986)).

numbers	measures	P. tongxinense sp. nov.	P. gracile	P. fahlbuschi	P. platyodon	
1	Distance between occipital condyle and premaxilla	584	–	–	–	
2	Distance between occipital condyle and nasal tip	598	590–598	516	550	
3	Distance between nasal tip and occipital crest	582	–	–	–	
4	Distance between nasal tip and bottom of nasal notch	186	184–191	185	215	
5	Minimal width of braincase	56	–	–	–	
6	Distance between occipital crest and postorbital process	297	–	–	–	
7	Distance between occipital crest and supraorbital process	329	–	–	–	
8	Distance between occipital crest and lacrimal tubercle	376	–	–	–	
9	Distance between nasal notch and orbit	67	66–67	43–35	58	
13	Distance between occipital condyle and M3	268				
14	Distance between nasal tip and orbit	243	–	–	–	
15	Width of occipital crest	68	–	–	–	
16	Width of paramastoid process	134	–	–	–	
17	Minimal width between parietal crests	35	–	–	–	
18	Width between postorbital processes	124	–	–	–	
19	Width between supraorbital processes	139	182–186	175	178	
20	Width between lacrimal tubercles	126	–	–	–	
21	Maximal width between zygomatic arches	134	258–265	270	205	
22	Width of nasal base	87	85–87	90–100	100	
23	Height of occipital surface	149	–	–	–	
25	Cranial height in front of P2	145	–	–	–	
26	Cranial height in front of M1	176	–	–	–	
27	Cranial height in front of M3	189	–	–	–	
28	Width of palate in front of P2	40	–	–	–	
29	Width of palate in front of M1	35	–	–	–	
30	Width of palate in front of M3	41	–	–	–	
31	Width of foramen magnum	38	–	–	–	
32	Width between exterior edges of occipital condyle	97	–	–	–	

Table 3 Measurements of the mandible of Plesiaceratherium tongxinense sp. nov. (IVPP V 23959).

(mm) (Measurements of P. gracile, P. fahlbuschi, P. platyodon, and P. mirallesi are from Yan & Heissig (1986)).

numbers	measures	P. tongxinense sp. nov.	P. gracile	P. fahlbuschi	P. platyodon	
1	Distance between anterior end of symphysis and mandible angle	470	433–435	435	455	
2	Distance between posterior end of symphysis and mandible angle	317	–	–	–	
3	Height of horizontal part of ramus at posterior edge of p2	61	–	–	–	
4	Height of horizontal part of ramus at posterior edge of p3	65	–	–	–	
5	Height of horizontal part of ramus at posterior edge of p4	68	–	–	–	
6	Height of horizontal part of ramus at posterior edge of m1	80	–	–	–	
7	Height of horizontal part of ramus at posterior edge of m2	89	–	–	–	
8	Height of horizontal part of ramus at posterior edge of m3	98	79–88	85–77	85	
9	Transverse diameter of lingual edge of horizontal part of ramus at p4-m1	52	–	–	–	
10	Transverse diameter of lingual edge of horizontal part of ramus at anterior edge of m3	61	19–21	35	34	
11	Length of symphysis along median plane	153	89–96	80–108	96	
13	Antero-posterior diameter of vertical part of ramus	131	–	–	–	
14	Transverse diameter of condyle	88	–	–	–	
15	Height of condyle	238	–	–	–	
16	Height of coronoid process	279	–	–	–	

Table 4 Comparisons between Plesiaceratherium tongxinense sp. nov. (IVPP V 23959) and other species of the genus Plesiaceratherium (after Mermier, 1895; Répelin, 1917; Wang, 1928; Young, 1937; Heissig, 1972; Antunes & Ginsburg, 1983; Ginsburg & Bulot, 1984; Yan & Heissig, 1986; Antoine, 2002; Becker & Tissier, 2020).

Species/Characters	P. tongxinense sp. nov.	P. gracile	P. platyodon	P. fahlbuschi	P. lumiarense	P. aquitanicum	P. balkanicum	
Frontoparietal crests	Closed	Sagittal crest	–	Closed	–	–	–	
Distance between the posterior edge of the nasal notch and the orbit	67 mm	66–67 mm	58 mm	43–35 mm	–	–	–	
The relationship between protoloph and ectoloph on P2	connected	connected	–	connected	connected	separated	separated	
Premolars	semi-molarized	semi-molarized	semi-molarized	molarized	semi-molarized	molarized	molarized	
crista on the upper molars	absent	present	–	absent	absent	present	absent	
crochet on the upper molars	strong	strong	weak or absent	strong	strong	strong	strong	
constrictions of the protocone and hypocone on the upper molars	strong	weak	–	strong	strong	–	strong	
the outline of M3 in occlusal view	quadrangular	–	triangular	quadrangular	quadrangular	triangular	quadrangular	
Cheek teeth	developed cement	without cement	without cement	without cement	without cement	without cement	without cement	
Distance between anterior end of symphysis and mandible angle	470 mm	433–435 mm	455 mm	435 mm	–	–	–	
Length of symphysis along median plane	153 mm	89–96 mm	96 mm	80–108 mm	–	–	–	

Description

Skull. The skull of IVPP V 23559 is complete and well-preserved with the upper cheek teeth moderately worn. The skull is slightly deformed by lateral compression, with the basioccipital and basal pterygoid parts narrowed, and the palatal bones deeply sunken while keeping the tooth rows close together.

In lateral view, the dorsal skull profile is flat and long. The occipital part is slightly raised. The occipital condyle is low and small. The posttympanic process is short, fused with the paraoccipital process, and anteriorly touches the postglenoid process. The external auditory pseudomeatus is closed ventrally, and its proximal edge is short and located in the lower half of the occipital crest. The area between the temporal and occipital crests is depressed. The zygomatic arch is thin (particularly the middle part), the anterior end of which is located at the level of M1 and close to the cheek teeth row, and the posterior end of the dorsal edge has a short process. The temporal condyle is articulated with the mandible protruding from the ventral edge of the zygomatic arch. The postglenoid process is laterally flattened. The position of the dorsal margin of the orbit is high, and the anterior margin of the orbit is located at the level of the M1/M2 boundary. The supraorbital edge of the frontal bone has a coarse surface but lacks any process or tubercle. The posterior orbital border is formed by the zygomatic bone, and presents a coarse area, without any tubercles. The nasal bone is thin and flat without lateral apophyses on both sides. The nasal notch has a U-shaped outline, and its posterior edge is at the level of the middle part of P4. The distance between the posterior edge of the nasal notch and the orbit is short, about 67 mm. The infraorbital foramen is located dorsally to the level of P4 and posteriorly to the nasal notch. The premaxillary bones are well-preserved with heavily worn I1s.

In dorsal view, the parietal crests are not fused to a sagittal crest, and the smallest width between parietal crests is located anterior to the nuchal crest, about 35 mm. The frontals are constricted at the middle of the temporal fossa. The ratio of zygomatic width to frontal width is greater than 1.5. The postorbital process is present. The widest position of the dorsal surface is located between the supraorbital processes, about 139 mm. The nasal bone narrows gradually before the orbits (i.e., the nasal base does not have a constriction). The nasal bone is narrow, flat, and long. A nasal suture is present.

In ventral view, the skull is long, with a length (from premaxilla to occipital condyle) of 584 mm. The ventral and occipital surfaces of the occipital condyle are rounded, without a median ridge. The hypoglossal foramen is laterally positioned, at the basement of the paraoccipital process. The alar foramen is opened on the lateral wall of the posterior nares, anteroposteriorly at the level of the temporal condyle. The tympanic bulla has been crushed, exposing the inner bones. The temporal condyle is high, and its transverse axis is straight. The posterior margins of the pterygoid are nearly vertical. The anterior edge of the posterior nares is V-shaped in outline, at the level of M3. The posterior edge of the lateral wall of the posterior nares with a steep part is continuous, extending to the foramen lacerum anterius that is at the back of the level of the temporal condyle. The medial and lateral edges of the cheek tooth row are nearly straight. The slender and straight premaxillary bones are two separated and faintly paralleled plates with a length of 66.7 mm. The I1 is deeply abraded and oval-shaped.

In posterior view, the outline of the occipital plate is bell-shaped. The occipital crest is rounded above and gradually inclined laterally. The nuchal tuberosity is developed. The foramen magnum is small, rounded in shape, and its width is 38 mm. The occipital condyles are relatively small, and their lateral margins have a short and steep upper part and a long and curved lower part. The width between exterior edges of occipital condyles is 97 mm.

Upper teeth. The upper cheek teeth have an undulate buccal wall, a developed expansion of the lingual cusps, an anteroposteriorly constricted protocone, absent enamel foldings, and buccal walls of cheek teeth covered by cement. The premolars are semi-molarized and have continuous lingual cingulum, closed medisinus, constricted protocone with curved lingual margin, and closed postfossette. The molars have developed antecrochet, developed crochet, absent crista as well as buccal cingulum, present lingual cingulum, and strongly constricted protocone and hypocone.

The I1 is oval shaped in the middle size with a longitudinal length of 26.5 mm in the abraded surface.

The DP1 is not preserved.

The P2 is nearly quadrangular in occlusal view with a relatively flat buccal wall. The parastyle fold and the paracone rib are weak. The protocone and hypocone connect by a lingual bridge. The hypocone is marginally larger than the protocone. The hypocone is at the same level as that of the metacone. The protoloph is as buccally narrow as the metaloph and joins with the ectoloph. The crochet and crista are connected forming a medifossette. Both the medisinus and the postfossette are closed. The anterior and the posterior cingula are developed. The lingual cingulum is V-shaped at the entrance of the medisinus.

The P3 has a weak parastyle fold and paracone rib with a shallowly undulating buccal wall. The hypocone has an anterior constriction. The semi-molarized P3 has developed crochet and crista, narrow and closed medisinus, small postfossette, and continuous lingual cingulum.

The P4 has a similar tooth structure but much larger than the P3. The former also has an expanded hypocone with an anterior constriction, a curved lingual margin of the protocone, and a continuous lingual cingulum.

The M1 has a projected parastyle with an undulating buccal wall. The strongly constricted protocone has a flat lingual margin, and the hypocone has a strong anterior constriction. The antecrochet is strong and elongates to the entrance of the medisinus. The medisinus leans to the narrow rear. The postfossette is round and closed. The anterior cingulum is developed, and the lingual cingulum is reduced, forming a pillar around the entrance of the medisinus.

The M2 has a long parastyle, a developed parastyle fold, and a paracone rib with an undulating buccal wall. Both the protocone and hypocone show strong constrictions. The M2 has a well-developed crochet, and strongly developed antecrochet with the stout end extends to the entrance of the medisinus. The antecrochet and hypocone are not connected. Besides, the M2 has an open medisinus, an oval-shaped and closed postfossette. Finally, the M2 has well-developed anterior cingulum, and the lingual cingulum is reduced, forming a pillar around the entrance of the medisinus.

The M3 has a quadrangular outline in occlusal view. The parastyle is short and sharp. The protoloph is wide and transverse on its antero-lingual side. The protocone develops strong anterior and posterior constrictions. The well-developed crochet does not encircle a medifossette. The anterior cingulum is continuous and well-developed. The posterior and lingual cingula are reduced and in shape of a pillar.

Mandible: The mandible of IVPP V 23559 is well-preserved, with the lower cheek teeth moderately worn. The mandible was slightly deformed by lateral shearing compression. The right i2 is broken, and the left p2 is lost. The horizontal ramus is long and raised. The lower margin is concave under the cheek teeth, with an upturned mandibular symphysis. The length of the mandibular symphysis along the median plane is long, about 153 mm. The posterior border of the mandibular symphysis is situated at the level of the p3. The oval mental foramen is small and located in the lower half of the horizontal ramus at the level before p2. The ascending ramus is relatively high with a height of 278 mm at the coronoid process, and 238 mm at the condyloid process. The mandibular condyle is transversely extended with a width of 87 mm, corresponding to the length of the glenoid fossa of the skull. The medial end of the condyloid process is curved posteriorly. The lateral half of the condyle is slightly inclined anteriorly. The mandibular notch between the coronoid and condyloid processes is narrow and deep. The lower part of the coronoid process is wide anteroposteriorly, and the upper part above the condyloid process tapers gradually as it curves posteriorly, with a flat anterior margin and rounded posterior margin. The posterior margin of the ascending ramus is slightly posteriorly inclined. The mandibular angle is rounded forming an obtuse angle. On the lateral surface of the ascending ramus, the masseter fossa is very deep under the coronoid process. The medial surface of the ascending ramus is depressed. The mandibular foramen is very large and situated anteriorly, with its bottom above the alveolar level. The groove behind the mandibular foramen is deep and wide, extending upward.

Lower teeth: The lower teeth are moderately worn. The row of the lower cheek teeth is aligned with the longitudinal axis of the horizontal ramus.

The i2 is medium-sized, dagger-shaped, and extending obliquely forward and upward almost parallel, with a root thicker than the crown. Its transverse section is a round triangle with an interior sharp angle, and the cross section of the root is oval.

The p2 is small in a triangular shape. It has a short and wide protolophid and a shallow ectoflexid. The trigonid basin is small and open, and the talonid basin is rounded and nearly disappeared. The buccal cingulum is developed under the hypolophid but absent under the protolophid.

The p3 is trapezoid in the occlusal view, with a slightly shorter anterior margin than the posterior one. The postero-buccal corner of the protoconid is rounded. The ectoflexid is shallow. The metalophid is robust, much wider than the entolophid. The trigonid basin is small and shallowly V-shaped, and the talonid basin is deeply V-shaped. The lingual cingulum is developed, and the buccal cingulum is developed under the hypolophid but absent under the protolophid.

The p4 is similar to p3 in morphology, but bigger in size. The occlusal surface is nearly rectangular. The postero-buccal corner of the protoconid is more angular than that of p3. The ectoflexid is wider and deeper than that of p3. The trigonid basin is nearly disappeared, and the talonid basin is deeply V-shaped.

The m1 is deeply worn. The occlusal surface is nearly rectangular. The postero-buccal corner of the protoconid is nearly right-angled. The ectoflexid is wide and shallow. The trigonid basin is nearly disappeared, and the talonid basin is deeply V-shaped. The lingual cingulum is reduced, and the buccal cingulum is absent.

The m2 has a rectangular occlusal surface. The postero-buccal corner of the protoconid is right-angled. The ectoflexid is shallowly V-shaped. The metalophid is robust and wider than the entolophid. The trigonid basin is U-shaped and the talonid basin is deeply V-shaped. Both the protolophid and hypolophid are slightly lingually oblique. The buccal cingulum is absent, and the lingual cingulum is reduced.

The m3 has a trapezoid occlusal surface, with a slightly shorter anterior margin than the posterior one. The postero-buccal corner of the protoconid is right-angled. The ectoflexid is wide and shallow. The metalophid is robust and wider than the entolophid. The trigonid basin is nearly disappeared, and the talonid basin is deeply V-shaped. Both the protolophid and hypolophid are slightly oblique lingually. The buccal cingulum is absent, and the lingual cingulum is reduced.

Comparison and discussion

The well-preserved new materials (IVPP V 23959) from Tongxin, Ningxia have typical features easily recognizable as typical of aceratheriine (Heissig, 1989; Cerdeño, 1995), including a flat and long nasal bone with a retracted nasal notch; the posttympanic process fused with the paraoccipital process; the upturned mandibular symphysis with large and straight i2s; the constricted lingual cusps on the upper cheek teeth.

However, the Tongxin specimen differs from derived aceratheriines (e.g., Chilotherium, Acerorhinus) in the morphology of the skull and mandible, as well as the degree of specialization of the incisors and cheek teeth. The Tongxin specimen differs from Mesaceratherium living in Eurasia from the upper Oligocene to lower Miocene by the relatively smaller I1s, and more complex occlusal patterns of the upper cheek teeth (Heissig, 1969; Blanchon et al., 2018). The Tongxin specimen also differs from Alicornops by the relatively smaller I1s, shorter distance from the nasal notch to the orbit, relatively low nuchal crest above the parietal and frontal surfaces, the reduction of buccal and lingual cingulum of lower molars, and the presence of a medifossette on upper premolars and a longer crochet on upper molars (Cerdeño & Sánchez, 2000; Deng, 2004; Heissig, 2012). The Tongxin specimen with developed I1s is different from the derived Eurasian aceratheriines, such as Hoploaceratherium, Chilotherium, Acerorhinus, Subchilotherium, and Shansirhinus (Borissiak, 1915; Ringström, 1924; Colbert, 1935; Deng, 2005; Heissig, 2012).

The Tongxin specimen shares diagnostic characteristics with the genus Plesiaceratherium, such as the narrow and slightly raised nuchal crest; the posttympanic process anteriorly touches the postglenoid process; the ventrally closed pseudomeatus external auditory; the nasal notch retracted to the level of P4; the upturned mandibular symphysis with large and straight i2s; the medium-sized I1s; the constricted lingual cusps on the upper cheek teeth; the narrow and long lower premolars with relatively shallow ectoflexid. Therefore, based on the combination of characters, we refer the Tongxin specimen to the genus Plesiaceratherium (Young, 1937; Yan, 1983; Yan & Heissig, 1986).

Compared with P. gracile (Young, 1937; Lu et al., 2016), the Tongxin specimen differs by the relatively wide parietal crests, but those of P. gracile are fused to form a single sagittal crest. The constrictions of the protocone and hypocone on the upper molars of the Tongxin specimen are stronger than those of P. gracile. The antecrochet and crochet on the upper molars of the Tongxin specimen are developed and stout, whereas P. gracile has slightly developed antecrochet and slender crochet on the upper molars. The crista on the upper molars of the Tongxin specimen is absent, but that of P. gracile is present. The cheek teeth of the Tongxin specimen are covered by cement on the buccal walls different from that of P. gracile without cement.

The skull of Tongxin specimen is larger in size than P. fahlbuschi (Heissig, 1972). The distance between the posterior edge of the nasal notch and the orbit of the Tongxin specimen is longer than that of P. fahlbuschi, which respectively are about 67 mm and 43–35 mm. The parietal crests of the Tongxin specimen are relatively wide, but those of P. fahlbuschi are fused to form a single sagittal crest. The anterior margin of the orbit of the Tongxin specimen is retracted at the level of the M1/M2 boundary, and that of P. fahlbuschi is located at the level of middle part of M1. The Tongxin specimen has semi-molarized upper premolars with the protocone and hypocone connected by a lingual bridge, while P. fahlbuschi has molarized upper premolars with the protocone and hypocone separated.

Compared with P. platyodon, the skull of the Tongxin specimen is larger in size. The distance between the posterior edge of the nasal notch and the orbit of the Tongxin specimen is about 67 mm longer than that of P. platyodon (∼58 mm). The anterior margin of the orbit of the Tongxin specimen is retracted at the level of the M1/M2 boundary, and that of P. platyodon is located at the level of middle part of M1. The crochet on the upper molars of the Tongxin specimen is developed and stout, whereas that of P. platyodon is weak or absent. The M3 of the Tongxin specimen has a quadrangular outline in occlusal view, while that of P. platyodon has a triangular outline.

The preserved materials of other Plesiaceratherium species are scarce. Compared with P. lumiarense, the Tongxin specimen has semi-molarized upper premolars with the protocone and hypocone connected by a lingual bridge, while P. lumiarense has upper premolars with the protocone and hypocone mostly separated. The Tongxin specimen differs from P. aquitanicum by the following features: its protoloph joins with the ectoloph on P2 but that of P. aquitanicum is separated from the ectoloph; the crista on the upper molars of the Tongxin specimen is absent, but that of P. aquitanicum is present; the M3 of the Tongxin specimen has a quadrangular outline in occlusal view, while that of P. aquitanicum has a triangular outline. It differs from P. balkanicum in its semi-molarized upper premolars with the protocone and hypocone connected by a lingual bridge (molarized upper premolars with the protocone and hypocone separated in P. balkanicum). The protoloph of the Tongxin specimen joins with the ectoloph on P2 but that of P. balkanicum is separated from the ectoloph.

Therefore, the Tongxin specimen is distinguished from all known species of the genus Plesiaceratherium by the following combination of characters: the skull is long and relatively flat with separated parietal crests; the supraorbital margin is high and its anterior margin is located at the level of the M1/M2 boundary; the upper I1 is developed and specialized, medium in size, with an oval abraded surface; the upper premolars are semi-molarized with the protocone and hypocone connected by a lingual bridge; the protocone on the upper molars has developed anterior and posterior constrictions; the buccal cingulum is absent on upper cheek teeth; finally, the M3 has a quadrangular outline in occlusal view. In comparison with Plesiaceratherium sp. from Japan (Fukuchi & Kawai, 2011), the premolars of the Tongxin specimen are much bigger than those of Plesiaceratherium sp. Besides, the fossils referred to Plesiaceratherium sp. in Africa (Pickford, 1986) were incomplete and there are no photographs available, so we were unable to make additional comparisons. Based on the previous comparisons, we attribute the Tongxin specimen to a new species, P. tongxinense sp. nov.

Although retaining some primitive characters, Plesiaceratherium is already a rather specialized genus as exemplified by the complex occlusal surface of the upper cheek teeth, the rather deep nasal incision, and the ventrally closed pseudomeatus external auditory. The genus represents an earlier taxon within Aceratheriinae (Yan & Heissig, 1986). As far as the dentition is concerned, Aceratherium and Plesiaceratherium are almost indistinguishable, and the skull characters are also similar (Yan, 1983). Therefore, the study of postcranial remains is necessary to further understand the relationship between Aceratherium and Plesiaceratherium and to establish the phylogenetic position of Plesiaceratherium.

The genus Plesiaceratherium is widely distributed in Eurasia with various occurrences in China, South Asia, and Europe (Young, 1937; Yan & Heissig, 1986; Antoine, Bulot & Ginsburg, 2000; Peter, 2002; Antoine et al., 2010). The earliest representative of this genus is P. naricum, from the earliest Miocene of Pakistan (MN 1-MN 2) (Antoine et al., 2010; Antoine et al., 2013; Antoine & Becker, 2013). In Europe, Plesiaceratherium was previously discovered in six localities, Sandelzhausen and Voggersberg in Germany, Pont du Manne as well as Estrepouy in France, Charneca de Lumiar in Portugal, and Can Julia in Spain (Heissig, 1999; Antoine & Becker, 2013). In China, Plesiaceratherium was found in three localities, namely, Shanwang in Linqu, Shandong Province (Young, 1937; Yan & Heissig, 1986), Jiulongkou in Cixian, Hebei Province (Chen & Wu, 1976), and Lunbori in Baingoin, northern Tibet (Deng et al., 2012). According to published data, the age of the Shanwang Fauna was about 18 Ma (Deng, Wang & Yue, 2003; Deng et al., 2012). The fossil locality of the Jiulongkou Fauna is the latest Shanwangian age, at about 16 Ma (Deng et al., 2012). In addition, the upper part of the Dingqing Formation at the Lunbori locality bearing Plesiaceratherium fossil is characteristic of the Early Miocene (Deng et al., 2012). Moreover, the localities yielding the more progressed Plesiaceratherium in Europe belong to MN 4 or 5 of the mammalian ages at 18-15 Ma (Steininger, 1999; Deng et al., 2012). Thus, the localities in Plesiaceratherium’s distribution in Eurasia are very close in age, i.e., the late Early Miocene. The new fossil materials of Plesiaceratherium reported here were discovered in the lower part of the Zhang’enbao Formation exposed in Miaoerling, corresponding to the late Early Miocene (Wang et al., 2016). Therefore, P. tongxinense survived in the late Early Miocene.

Phylogenetic analysis

To explore the phylogenetic relationships of the Tongxin specimen, we performed a phylogenetic analysis of the Rhinocerotidae (based on the data matrix of Antoine (2002); Antoine (2003), with the addition of Plesiaceratherium tongxinense sp. nov., P. fahlbuschi, P. gracile, P. lumiarense, P. platyodon, Mesaceratherium welcommi, Mesaceratherium gaimersheimense, Alicornops simorrense, Hoploaceratherium tetradactylum, Chilotherium anderssoni, Chilotherium wimani, Aceratherium incisivum, Acerorhinus zernowi, Acerorhinus yuanmouensis, and Shansirhinus ringstroemi, resulting in four equally most parsimonious trees. The length of the four most parsimonious trees is 1,233 steps, with a consistency index of 0.3001 and a retention index of 0.5738.

All members of Teleoceratina are clustered in a single clade (Node A) (Fig. 4). They share thirty-one unequivocal synapomorphies including transversal profile of articular tubercle concave (ch. 40), processus postglenoidalis dihedron (ch. 42), posterior groove on the processus zygomaticus present (ch. 45), cement on cheekteeth abundant (ch. 66), shape of the crown on I1 almond (ch. 72), labial cingulum on upper premolars usually present (ch. 83), medifossette on P3-4 always absent (ch. 100), crista on P3 always absent (ch. 105), lingual opening of the posterior valley on lower premolars U-shaped (ch. 146), d1/p1 in adults usually absent (ch. 151), di1 absent (ch. 170), posterior valley on d2 usually open (ch. 180), scapula spatula-shaped (H/APD ≤ 1.5) (ch. 190), glenoid fossa of scapula medial border straight (ch. 191), proximal ulna-facets of radius usually fused (ch. 199), gutter for the m. extensor carpi of radius weak (ch. 202), postero-proximal facet with semilunate of scaphoid present (ch. 207), posterior tuberosity of magnum short (ch. 220), magnum-facet in anterior view of McIII invisible (ch. 229), fovea capitis of femur high and narrow (ch. 238), proximal border of the patellar trochlea of femur straight (ch. 241), antero-distal groove of tibia absent (ch. 242), medio-distal gutter (tendon m. tibialis posterior) of tibia usually present (ch. 243), 1.2 ≤ the ratio between transverse diameter and height of astragalus (ch. 252), posterior stop on the cuboid-facet of astragalus absent (ch. 257), calcaneus-facet 1 of astragalus nearly flat (ch. 262), fibula-facet of calcaneus usually present (ch. 264), tibia-facet of calcaneus always present (ch. 265), cuboid-facet of MtIII present (ch. 275), limbs robust (ch. 279), and the insertion of the m. interossei of lateral metapodials short (ch. 282).

The clade including the plesiaceratheres (Node B; Fig. 4) is supported by twenty unequivocal synapomorphies including contact of nasal and lachrymal long (ch. 6), dorsal profile of skull flat (ch. 15), skull dolichocephalic (ch. 23), nasal bones very long (ch. 26), vomer rounded (ch. 38), foramen magnum circular (ch. 49), foramen mentale in front of p2 (ch. 56), ramus inclined forward (ch. 60), processus coronoideus of ramus little developed (ch. 61), cheekteeth low crowned (ch. 68), lingual bridge between protocone and hypocone on P2 (ch. 94), crista on P3 always present (ch. 105), crista on upper molars usually absent (ch. 112), metaloph on M1 continuous (ch. 125), metaloph on M2 continuous (ch. 129), d1/p1 (in adults) always present (ch. 151), fossa olecrani of humerus high (ch. 193), distal border of anterior side of semilunate rounded (ch. 212), collum tali of astragalus low (ch. 256), distal widening of the diaphysis (in adults) of MtIII present (ch. 274). The new species Plesiaceratherium tongxinense established here is located in the basal position of the genus Plesiaceratherium supported by twenty-one unambigous synapomorphies including nuchal tubercle little developed (ch. 20), posterior margin of pterygoid nearly vertical (ch. 22), frontoparietal crests closed (ch. 35), articular tubercle of squamosal high (ch. 39), processus postglenoidalis dihedron (ch. 42), sagittal crest on the basilar process of basioccipital absent (ch. 44), processus posttympanicus and processus paraoccipitalis fused (ch. 46), base of mandibular corpus convex (ch. 59), ramus inclined backward (ch. 60), mandibular foramen above the teeth neck (ch. 62), the radio of compared length of the premolars/molars rows between 42% to 50% (ch. 63), metaloph constriction on P2-4 present (ch. 86), antecrochet on P2-3 usually present (ch. 90), medifossette on P3-4 always present (ch. 100), antecrochet on P4 usually present (ch. 107), labial cingulum on upper molars usually absent (ch. 109), crista on upper molars always absent (ch. 112), constriction of the protocone on M1-2 strong (ch. 116), crista on M1-2 always absent (ch. 123), shape of M3 quadrangular (ch. 134), posterior groove on the ectometaloph on M3 absent (ch. 138). P. tongxinense may be at an early stage, while the later flourishing of P. gracile and the diversity of European Plesiaceratherium indicate that the genus has reached an adaptive radiation stage. The original features of P. tongxinense include little developed nuchal tubercle, absent sagittal crest on the basilar process of basioccipital, fused processus posttympanicus and processus paraoccipitalis, always absent crista on upper molars, and quadrangular shape of M3.

Node C contains elasmotheres (Fig. 4) and is consistent with the results of phylogenetic analysis of Sun et al. (2023). Diceratherium armatum and Menoceras arikarense are the sister-groups of Elasmotheriina (i.e., elasmotheriines sensu stricto) which is consistent with the results of phylogenetic analysis of Antoine (2003) and Sun et al. (2023).

Conclusions

The morphology of the particularly complete skull and mandible from (IVPP V 23959) from Tongxin, Ningxia described here do not quite match those of any known aceratheres. Although the skull and mandible conform to the generic characters of Plesiaceratherium, they differ from all the species of this genus. Based on the combination of characteristics and phylogenetic analysis, we herein establish the new species as Plesiaceratherium tongxinense sp. nov.

The phylogenetic analysis reveals that P. tongxinense nov. sp. is located basally within Plesiaceratherium. Such placement is supported by these unequivocal synapomorphies: little developed nuchal tubercle, nearly vertical posterior margin of pterygoid, closed frontoparietal crests, high articular tubercle of squamosal, processus postglenoidalis dihedron, absent sagittal crest on the basilar process of basioccipital, fused processus posttympanicus and processus paraoccipitalis, convex base of mandibular corpus, inclined backward ramus, mandibular foramen above the teeth neck, the radio of compared length of the premolars/molars rows between 42% to 50%, present metaloph constriction on P2-4, usually present antecrochet on P2-3, always present medifossette on P3-4, usually present antecrochet on P4, usually absent labial cingulum on upper molars, always absent crista on upper molars, strong constriction of the protocone on M1-2, quadrangular shape of M3, and absent posterior groove on the ectometaloph on M3. The discovery of this new species not only improves the morphological characteristics but also increases the diversity of the plesiaceratheres.

Supplemental Information

Supplemental Information 1 Feature matrices

Click here for additional data file.

Supplemental Information 2 Synapomorphy list

Click here for additional data file.

We thank Prof. Zhanxiang Qiu for his constructive suggestions and comments. We thank Wei Gao for his photographs and Xiaocong Guo for her illustrations. We are grateful to the editors and reviewers for their constructive comments on the improvement of the manuscript.

Abbreviations

I/i upper/lower incisor

L length

M/m upper/lower molar

P/p upper/lower premolar

W width

MN European Neogene Mammal Zones

BSPG Bayerische Staatssammlung für Paläontologie und Geologie, Münich, Germany

IVPP Institute of Vertebrate Paleontology and Paleoanthropology, Chinese Academy of Sciences, Beijing, China

KNM Kenya National Museum, Nairobi, Kenya

MNHN Muséum National d’Histoire Naturelle, Paris, France

Additional Information and Declarations

Competing Interests

Author Contributions

Data Availability

New Species Registration

The authors declare there are no competing interests.

Danhui Sun conceived and designed the experiments, performed the experiments, analyzed the data, prepared figures and/or tables, authored or reviewed drafts of the article, and approved the final draft.

Tao Deng conceived and designed the experiments, performed the experiments, analyzed the data, authored or reviewed drafts of the article, and approved the final draft.

Shiqi Wang performed the experiments, analyzed the data, authored or reviewed drafts of the article, and approved the final draft.

The following information was supplied regarding data availability:

The feature matrices are available in the Supplemental File.

The following information was supplied regarding the registration of a newly described species:

Publication LSID: urn:lsid:zoobank.org:pub:D79AA940-F686-4CD7-A751-C94CC0E30E44

Plesiaceratherium plesiaceratherium tongxinense species LSID: urn:lsid:zoobank.org:act:C9581013-82FE-4AEF-9DA3-133AACBAE851

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
