# Peer review of "New materials of plesiacerathere (Perissodactyla, Rhinocerotidae) from the late Early Miocene of Northern China"

_PeerJ, doi:10.7717/peerj.16822_

## Round 0.1 · original submission · Major Revisions

Dear authors,

Having read the manuscript and the referees’ assessments, I agree with them that the main concerns about the present manuscript are:

- Grammar and spelling of the paper. Please, check it previous resubmission. See comment of referee 2 and his suggestions in the text.

- Comparison section: To improve the comparison section including a table describing the traits of the new species, P. gracile, and, if it is possible, other Plesiaceratherium species.

- Phylogenetic analysis: To consider to perform more phylogenetic analysis with more close species (e.g., Aceratheriina), including also those from African and Japanese regions (they are not mentioned now at the text).

- Discussion and conclusions: To discuss relevant issues relevant issues about the taxonomy and paleobiogeography of the group. See comments of the two referees.

- Supplementary: To allow the replicability of your analysis, please include tnt file as supplementary in further submissions.

Please, provide a new version of the manuscript considering these major concerns and also other comments and advice that reviewers have raised in their review or annotated manuscripts.

Thanks in advance, awaiting your submission.

Blanca Moncunill-Solé

**Language Note:** The Academic Editor has identified that the English language must be improved. PeerJ can provide language editing services - please contact us at [email protected] for pricing (be sure to provide your manuscript number and title). Alternatively, you should make your own arrangements to improve the language quality and provide details in your response letter. – PeerJ Staff

·

Basic reporting

This result will contribute to the future descussion of phylogenetic relationship and paleobiogeography Eurasia of Plesiaceratherium.
The grammar of the manuscript is appropriate enough, and the figures are adequate.
The raw data for the cladistic analysis have been properly added as supplimentary failes.
Regarding to the overview of the fossil records of Plesiaceratherium, I recommend to add author's opinion for the African and Japanese plesiaceratheres. Beacause a few plesiaceratheres have been found from these areas.
For figure 4, I recommend to add the explain of the numbers near the each nodes.

Experimental design

The authors note that the purpose of this manuscript is describing the new species of Plesiacerathereium from northen China and to discuss the distribution of this taxa in Eurasia and stratigraphic implication.

I think this study has well verieid the comparison and analyzed the phylogeny to discuss the above purpose, but I hope some additional explanations for the taxonomic framework of the Subfamily treated in this study and several characters of the studied specimen.

Validity of the findings

Previously, the fossil records of Plesiaceratherium in Asia have been discontiuned,in contrast to those of Europe have been well known.
This manuscript reviews the prebiously known Plesiaceratherium and stratigraphic ranges of them to discuss the occurance range of this genus.
Using the standard method, this study investigates the phylogenetic analysis for the species of Plesiaceratherium including the studied material.
In this study, the authors used the modified data matrix of Antoine (2002, 2003).
This is in my opinion, but I recommend to add the author's opinion for the recent phylogenetic analysis of the Aceratheriinae by Lu et al. (2023).

Additional comments

no comment

·

Basic reporting

The paper "A new plesiacerathere (Perissodactyla, Rhinocerotidae) from the late Early Miocene of northern China " by Sun and colleagues poses a relevant contribution to the field.

Overall, this study provides an interesting contribution to the rhinoceros systematics of this fairly large but somewhat poorly known genus. The literature is up to date, the authors show a comprehensive knowledge of the topic, and the conclusions, although partial, are relevant to the main goal of the article. The figures are outstanding, although the illustrated versions would benefit from a better contrast. Tables and raw phylogenetic data are also shared. However, as the main analyses have been conducted with TNT, I strongly recommend to include the tnt format file together with the nexus one.

There are more important issues that should be addressed in the current form of the manuscript. Most of them are detailed in detail in the following sections and the attached pdf file.

It is important to mention that there are some underlying issues with the English language that should be addressed before its publication. These are scattered through the manuscript and too widespread to be mentioned in detail. The most obvious ones have been detailed in the attached pdf document. Therefore, I strongly recommend a thorough revision by a native speaker.

Experimental design

The research questions are not defined in a traditional sense, as the authors have opted for a classic systematic approach. That said, the paper would benefit from a clearer statement of the main research questions, which are indeed interesting: this is a basal species found at the MN5, quite late for the . This particularity by itself is worth at least exploring in the conclusion section.

The listed methods are detailed sufficiently and can be replicated easily.
There is a potential flaw in the phylogenetic analysis proposed by the authors. Even though Plesiaceratherium has been placed within Aceratheriina (as the authors mention), the species selected for the phylogenetic analysis are mostly Elasmotheriinae, a Subfamily unrelated with the former. The phylogenetic framework would enormously benefit from a tighter taxa selection. This could be solved by including more Aceratheriina, particularly basal species, and some Rhinocerotina as well. That would clarify the phylogenetic position of this genus from a wider perspective and widen the systematic scope of the manuscript, right now focused on the position of the species within the genus Plesiaceratherium.

There is also a preliminary attempt to sort out the paleobiogeographic origin of the group (by coloring the species names in figure 4, but little mention to the paleobiogeographic context or how they influence the particular stratigraphic range of the species is outlined in the conclusions section.

Validity of the findings

According to the phylogenetic analysis performed by the authors, the form from Miaoerling has a unique set of synapomorphies that justify its ascription as a new species. Superficially, the potentially new taxon is somewhat similar to Pl. gracile and other European taxa as well. A comparison table outlining the main differences between these two species (ideally, other closer ones) would greatly help future specialists trying to distinguish between them. This could be also represented as an additional figure comparing these species visually.

The conclusions are succinct, and some of the questions that automatically arises from this basal species (how is it different from other Asian species? What are the hypotheses behind the late age of the species in combination with its basal condition? Is there any relevant paleobiogeographic information behind those statements?) remain largely unsolved despite being mentioned through the manuscript its key placement filling the discontinuous distribution of the genus.

---

## Round 0.2 · Minor Revisions

Dear authors,

First of all, I would like to apologize for the delay in the reviewing process of your paper. We had several problems with one of the referees, who has not send in time his/her review. So, this have made that the reviewing process has been so long. Sorry again.

Secondly, considering the referees comments in this revised version of the manuscript, I agree with them that previous acceptation some minor changes should been done to have strengther results and conclusions of your study, specially writing mistakes (grammar, spelling, etc.). Pease, check carefully the reviewers 1’ comments, as well as the document attached of referee 2 where you can find specific changes (I am attaching it as editor).

Please, provide a new version of the manuscript considering these concerns.
I want to wish Merry Christmas, and a happy and productive new year.
Thanks in advance,
Blanca

·

Basic reporting

no comment

Experimental design

Comment to the method;

In this study, the pylogenetic analysis has been carried out using the data of Antoine (2002, 2003) to invetigate of the Aceratheriinae.

Recently, Lu et al. (2023) analyzed the phylogenetic framework of the Aceratheriinae using their new data.
What opinion dou you have for that study?
If it possible, i recommend to aad the reason for using Antoine's one.

ref)Lu, X.-K.; Deng, T.; Pandolfi, L. Reconstructing the Phylogeny of the Hornless Rhinoceros Aceratheriinae. Front. Ecol. Evol. 2023, 11, 1005126. https://doi.org/10.3389/fevo.2023.1005126.

Validity of the findings

no comment

Additional comments

no comment

·

Basic reporting

Annotated manuscript is attached.

Experimental design

Annotated manuscript is attached.

Validity of the findings

Annotated manuscript is attached.

---

## Round 0.3 · accepted · Accept

Dear authors,
Congratulations! In this final version, you have addressed all the referees' comments successfully. From my point of view, this new version has a very clear guiding thread, and all the sections are clear and well-explained. When reading it, however, I have noted some minor typographical errors. I detail them below. Please, correct them in the fully typeset publication proofs, together with others that you can identify in them, before final publication.
I wish you a Happy New Year,
Cheers,

PhD Blanca Moncunill-Solé

Lines 64-66. Antunes & Ginsburg (1983) established.... which was previously identified as Aceratherium lumiarense by Antunes & Ginsbutg (1983). It is the same reference. I think that there is mistake in the year.
Line 87. Include a blank space between "nov." and "providing"
Line 119. Delete quotation mark at the end of the sentence (").
Line 146. Include a dot "." at the end of the sentence.
Line 166. Delete one of the blank spaces that follows "boundary;".
Line 218. Should "lacerum anterius" be in italics?
Line 276. Include a blank space between "antecrochet" and "with".
Line 334. Replace ":" that follows "in size" by a full stop ".".
Line 371. Replace "Sanchez" by "Sánchez".
Line 448. Delete on e of the blank spaces that are between "2011,)" and "the premolars".
Line 653. Replace "Sanchez" by "Sánchez".